# Barriers to Change: Social Network Interactions Not Sufficient for Diffusion of High-Impact Practices in STEM Teaching

Tracie Reding [1], Christopher Moore [2,*], Julie A. Pelton [3] and Sarah Edwards [4]

1   Parlay Consulting Firm, Omaha, NE 68182, USA; traciereding@parlayconsultingfirm.com
2   Physics Department, University of Nebraksa Omaha, Omaha, NE 68182, USA
3   Sociology & Anthropology Department, University of Nebraska Omaha, Omaha, NE 68182, USA; jpelton@unomaha.edu
4   Teacher Education Department, University of Nebraksa Omaha, Omaha, NE 68182, USA; skedwards@unomaha.edu
*   Correspondence: jcmoore@unomaha.edu; Tel.: +1-402-554-5981

**Abstract:** We examined the relationship between faculty teaching networks, which can aid with the implementation of didactic high-impact practices (HIPs) in classroom instruction, and the actual implementation of said practices. Participants consisted of STEM faculty members that teach introductory courses at a USA research university. A total of 210 faculty were invited to complete the Teaching Practices Inventory (TPI), which measures the use of classroom-based HIPs, and were then directed to a follow-up survey to gather teaching network data if they qualified. A total of 90 faculty completed the TPI, with 52 respondents completing the network analysis portion. Ego-level data, as well as network structural position data, were collected through roster format listing all invited faculty. No correlations were found between these network metrics and TPI score. Furthermore, respondents with similar TPI scores showed no preference for interactions within their group. For example, faculty with widely varying TPI scores interacted with each other with no indications of HIPs diffusion. Although the literature suggests strong teaching networks are a necessary condition for broad diffusion of HIPs, these results indicate that such networks are not a sufficient condition. This has implications for the diffusion of HIPs specifically and institutional change generally. Engaging individuals that possess both structural positions and pedagogical knowledge may be needed to help strategically diffuse HIPs in their own networks, with institutional support and guidance most likely also required.

**Keywords:** high-impact practices; teaching practices; social networking analysis; teaching networks; teaching practices inventory; institutional change; instructional change; concerns-based adoption

## 1. Introduction

Efforts toward educational reform in higher education STEM courses have consistently increased over the last several years. These efforts are bolstered by the numerous federal policies and grants supporting research and implementation of high-impact practices [1]. Classroom-based high-impact practices (HIPs) are educational practices deployed at the individual instructor level that show academic benefits for all students, including those from under-served populations [2]. As opposed to institutional high-impact practices, the ability for classroom-based HIPs to impact student success on a large scale relies on widespread adoption and fidelity of implementation by faculty members, which remain challenges within STEM courses in American universities [3].

While widespread adoption and implementation fidelity are known challenges, there are some factors that have been shown to support changes in teaching. For example, the formation of strong social networks has been shown to facilitate education reform within the K–12 environment [4], where educator social networks have led to the diffusion of HIPs throughout K–12 departments, schools, and districts [4–6]. Although viewing



education reform through a social network lens is prevalent in K–12 settings, less work has been situated within higher education, though this is changing. Recent studies with post-secondary institutions have shown that faculty social networks help create an environment conducive to the diffusion and implementation of HIPs [7–11].

Within these studies, most focus on a single department instead of an entire campus and have identified the need for expanding the networks outside of singular departments [7,11]. This study encompasses multiple departments across a single campus to shed light on inter- and intra-department teaching networks with the aim of assisting campus-wide leaders to plan and implement pedagogical changes.

Social network analysis (SNA) has seen increasing use within post-secondary institutions, with the intent to help plan and implement changes in practice. However, this necessitates increased research to develop the knowledge base to use SNA effectively towards institutional change. This article investigates the relationship between the prevalence of HIPs in science, technology, engineering, and mathematics (STEM) courses and the social networks created by the faculty teaching these courses.

The research question we seek to answer in this article is as follows: "Do faculty interactions about teaching correlate with greater use of HIPs?" Specifically, previous work has shown strong faculty social networks to be a necessary condition for institutional change within units in higher education [7–11]. However, we address whether or not such networks are a sufficient condition.

## 2. Background

### 2.1. High-Impact Teaching Practices

The term "High Impact Practices" (HIPs) may garner specific thoughts and notions of what these practices might look like in the classroom with many methods of assessing their implementation including self-report surveys and observation protocols. For purposes of determining the extent to which faculty members implement HIPs, this research relied on the use of the self-report survey known as the Teaching Practices Inventory (TPI) [12]. The TPI was used in this study because it provided a general enough framework to be applied across different STEM disciplines while being specific enough to drive meaningful action which is necessary to influence the implementation of HIPs [13].

Designed to characterize the teaching practices used in undergraduate science and mathematics courses, the TPI is a survey administered to instructors that requires 10 min or less to complete. The TPI provides a detailed characterization of the practices used in all aspects of lecture-based courses. As described by the authors of the TPI, "the results allow meaningful comparisons of the teaching used for the different courses and instructors within a department and across different departments" [12]. The survey is not suitable for use with courses that are primarily laboratories, seminars, or project courses, such as course-based undergraduate research experiences (CUREs). The survey has been validated and shown to be reliable through use at multiple different universities and has been modified for validity to social science courses [12,14].

Mean TPI scores can also be analyzed by category of practice, further defining support needs. Analysis of individual categories can help decision-makers determine which areas to target support and resources, ensuring allocation where most needed [15]. For example, a unit-level report of comparatively low scores within the "Assignments" category could indicate the necessity of an intervention targeted to that specific category, such as workshops led by a campus teaching center, department chair discussions with faculty in department meetings, etc. This allows for the interventions to target identified needs, as opposed to more general "scattershot" approaches to intervention [15].

The general factors and practices that have been shown to support learning across a wide range of STEM disciplines include knowledge organization, reducing cognitive load, motivation, metacognition, and group learning [12,16]. Table 1 shows a list of practices that support learning within each factor, with more detail provided in Ref. [12]. Along with factors that support learning, Table 2 shows general factors and practices that have been

shown to support teacher effectiveness, such as prior knowledge and beliefs, effectiveness, and gaining relevant knowledge and skills. These factors have also been shown to support learning and teaching effectiveness in social science disciplines, with the practices inventory showing validity across the natural and physical sciences, mathematics, engineering, and the social sciences with only minor changes to terminology [14].

**Table 1.** General Factors and Descriptions of Practices Shown to Support Learning. Modified from Ref. [12] in accordance with Creative Commons BY-NC-SA 3.0.

| Factor | Practices That Support Learning |
|---|---|
| Knowledge Organization | Provides a list of topics to be covered. <br> Provides a list of topic-specific competencies. <br> Provides a list of crosscutting competencies (problem-solving, etc.). <br> Provides out-of-class multimedia content. <br> Provides lecture notes or other class materials. <br> Spends course time on the process. |
| Reducing Cognitive Load | Provides worked examples. <br> Pre-class materials are provided. <br> Students read/view and are formatively quizzed before class time. |
| Motivation | The course attempts to change student attitudes and perceptions. <br> Articles from the scientific literature are used in the course. <br> Students discuss why the material is useful. <br> Students are explicitly encouraged to meet with the instructor. <br> Students are provided with opportunities to have some control over their learning. |
| Practice | Practice exams are provided. <br> Small-group discussions or problem-solving. <br> Demonstrations require students to first make predictions. <br> Student presentations are assigned. <br> A significant fraction of class time is spent not lecturing. <br> Peer-response systems are used, such as "clickers." <br> A paper or project is assigned involving some degree of student control. <br> A significant fraction of an exam grade requires reasoning explanation. |
| Feedback | Student wikis or discussion boards are used with instructor feedback. <br> Solutions to homework assignments are provided. <br> Solutions to exams are provided. <br> Instructor pauses to ask for questions. <br> Feedback is provided on assignments with opportunities for students to redo work. |
| Metacognition | Class ends with a reflective activity. <br> There are opportunities for self-evaluation. |
| Group Learning | Students are encouraged to work collaboratively on assignments. <br> There are explicit group assignments. |

### 2.2. Faculty Teaching Networks

Despite a growing awareness of the benefits of HIPs on undergraduate student success, instructors tend to rely on anecdotes and experience over empirical evidence for decisions regarding teaching practices [17]. The decision-making process of what and how to teach through anecdotes and experience is informed not only by the individual instructor's experiences but also by their peers' experiences [17,18]. This peer influence regarding teaching practices has been consistently demonstrated within the K–12 educational setting but is an emerging research focus in higher education [4,10]. According to McConnell et al., peers influence one another's instructional decision-making process in three ways: (1) sharing information, (2) reinforcing or changing attitudes, and (3) shaping and communicating teaching climate [19]. At the departmental level, teaching climate is defined as "an emergent property of a department's prevailing culture, disciplinary history, interactions between members of the department, and outside influences such

as institutional context and external stakeholders" [19]. All three ways of instructional practice influence are interdependent and are rooted in faculty member interactions.

**Table 2.** General factors and descriptions of practices shown to support teacher effectiveness. Modified from Ref. [12] in accordance with Creative Commons BY-NC-SA 3.0.

| Factor | Practices That Support Teacher Effectiveness |
| --- | --- |
| Prior Knowledge and Beliefs | Assessment of student knowledge and/or beliefs is done at the beginning of the course. A pre-post survey of student interests and/or perceptions is assigned. |
| Feedback on Effectiveness | Students complete a midterm course evaluation. The instructor repeatedly gains feedback from students. An instructor-independent pre/post-test is used to measure learning. New teaching methods are evaluated using measurements of the impact on learning. |
| Gain Knowledge and Skills | The instructor uses "departmental" course materials. The instructor discusses how to teach the course with colleagues. The instructor reads the literature about teaching and learning relevant to the course. The instructor sits in on a colleague's class. |

These faculty member interactions are a significant component of gaining abilities in implementing HIPs [16,20]. Social interactions like those between faculty members regarding teaching practices can be empirically studied through SNA. SNA seeks to measure interactions among network members and is an emerging approach to addressing network questions in various sectors, including higher education [21–27]. SNA can be used to investigate networks at multiple levels including ego, subgroup, and whole network. Metrics at the ego level help gain insights into individual members' propensity to diffuse advice throughout the network [28]. Subgroup metrics can help identify intra-network clusters and their properties [29]. Whole-network metrics can be used to elucidate the overall structure of the network including the percentage of connections that are realized, ease of communication among the members, and homophily [30].

These metrics can help facilitate reform efforts in various ways such as the determination of key actors within a group [27,31,32]. Future evidence-based intervention necessitates the identification of key institutional change agents that can serve as exemplars of the adoption of HIPs. Ego-level centrality measures, as well as whole-network metrics, can be used to inform campus leaders about the health of the instructor network, and the types, and strengths of interactions concerning teaching happening between instructors [30]. For example, SNA metrics and TPI scores could be used to identify emergent faculty leaders that could possibly assist with the diffusion of HIPs throughout introductory STEM courses.

### 2.3. Study Context and Objectives

This study is situated within a larger project seeking to understand the prevalence of and barriers to implementing HIPs in STEM courses. We adopt the USA Congressional Research Service's definition of STEM fields to include "mathematics, natural sciences, engineering, computer and information sciences, and the social and behavioral sciences—psychology, economics, sociology, and political science" [33], and therefore include social science course instructors within our studied population. The larger project also involves the implementation of a Concerns-Based Adoption Model (CBAM) that enables leaders to gauge faculty concerns and program use to give necessary support to ensure success.

As shown in Figure 1, there are three dimensions of the CBAM: the Model of Success (called Innovation Configuration Map), Measures of Behaviors, and Measures of Attitudes. The model of success used in this project is defined by the eight categories outlined in and measured with the TPI, which we are collectively referring to as HIPs [12]. The measures of behaviors include faculty self-report via the TPI survey, analysis of collaboration and sharing via SNA, and validation of self-reports using the Classroom Observation Protocol for Undergraduate STEM (COPUS) [31,32,34,35]. The measures of attitudes include the

development and deployment of an attitudinal survey based on the Stages of Concern Questionnaire (SoCQ), and faculty focus groups and interviews on structural barriers to the implementation of HIPs [36].

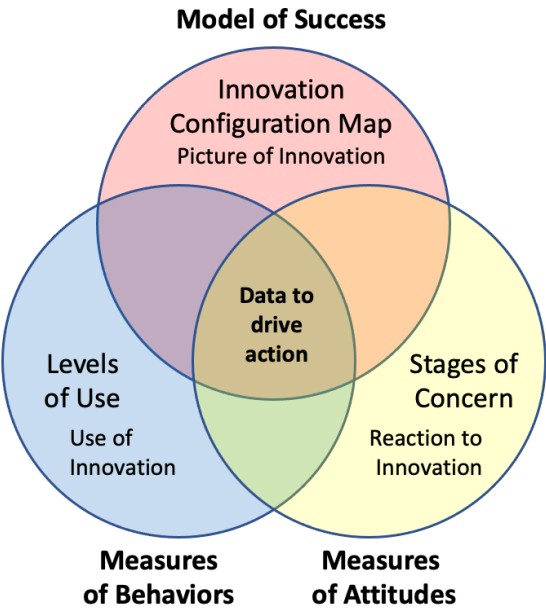

**Figure 1.** Schematic representation of the Concerns-Based Adoption Model (CBAM).

As a subset of the larger project, the work presented in this report investigates the relationship between measures of behaviors, specifically faculty self-reports on the TPI, and collaboration and sharing as measured by faculty member interactions regarding teaching through the SNA metrics previously discussed. The overarching research question for this study based on these components of the CBAM is as follows: How does a faculty member's implementation of HIPs relate to their teaching network?

Since the literature is clear that these faculty member interactions are a necessary component of gaining abilities in implementing HIPs, we hypothesized that there should be a significant relationship between HIP use and teaching network positioning [16,20]. It has been shown that anecdotes and experience within a faculty network drive the decision-making process of what and how to teach [4,10,17,18]. Peer influence regarding teaching practices has been consistently demonstrated to happen through sharing of information, reinforcing or changing of attitudes, and shaping and communicating teaching climate, all three of which are rooted in faculty member interactions measured by SNA metrics [4,10,19]. Based on this previous work, we, therefore, had the following hypotheses related to the larger research question:

**H1.** *Respondents with higher TPI scores will demonstrate higher ego metrics.*

**H2.** *Respondents with similar TPI scores will more frequently interact with one another than with individuals with different TPI scores.*

### 3. Methods and Results

This study employed a mixed methods social network analysis design by combining self-report TPI survey results with network analysis results. The initial study design focused on the combination of TPI results and a roster format SNA survey. The sampling procedures, data collection methods, and data analysis process are described below.

### 3.1. Data Collection

At the end of the fall 2020 academic semester, the TPI was distributed to 210 faculty members teaching general education science, mathematics, or social science courses at a Midwestern university using the cloud-based Qualtrics survey platform. The inventory was started by 119 respondents and completed by 90, resulting in a total response rate of 42.9%. We report all TPI scores as a total score out of the 72-point maximum score using the scoring methodology reported in Wieman's 2014 article [12].

If respondents indicated on the TPI that they had discussed teaching the course with others, then they were automatically provided a follow-up survey on the individuals with whom they have collaborated, called alters, and the depth of those interactions. Of the 90 respondents, 79 indicated some level of discussion about teaching occurred with other instructors, with 52 of those respondents completing the follow-up SNA survey. The follow-up survey consisted of a roster of all individuals from the total 210 initial member participant list broken down by college and department. Respondents were first asked if they had teaching discussions with anyone from a list of academic units. Respondents selecting a unit would then be shown a roster of individuals within that unit and asked with whom those teaching discussions occurred to identify alters. Finally, each survey respondent identified the level of interaction they had with each alter included in the roster. Interaction levels included the following prompts:

1. We have not interacted regarding teaching.
2. We have discussed teaching this course or a related general education course.
3. We have actively collaborated on this course or a related course, such as developing shared lessons or aligning curriculum.
4. We have worked together on the scholarship of teaching this course or a related course, such as a presentation, publication, or grant proposal.

### 3.2. Results and Analysis

Table 3 shows the mean and standard deviation for both the larger total TPI respondents and the smaller sub-set of respondents that also completed the SNA survey. Figure 2 shows the histogram of TPI scores for all respondents (light blue) and the respondents that both indicated teaching discussions on the TPI and self-selected to participate in the SNA survey (light green). For both groups, TPI scores were normally distributed.

**Table 3.** TPI score descriptive statistics.

| Group | *N* | Mean | S.D. |
|---|---|---|---|
| All Respondents | 90 | 33.2 | 8.2 |
| SNA Respondents | 52 | 35.4 | 7.7 |

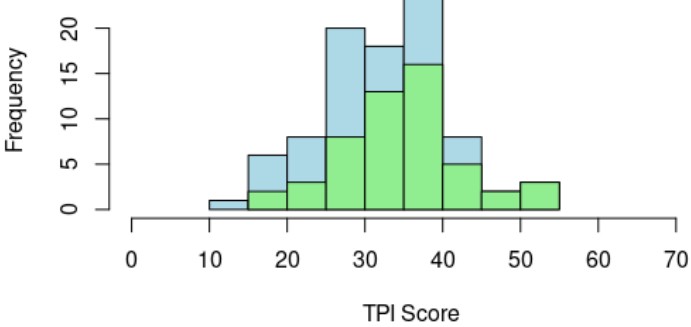

**Figure 2.** Histogram of TPI scores showing all respondents (light blue, $N = 90$) and respondents that also participated in the SNA component of the survey (light green, $N = 52$).

Initially, respondent interactions had to be verified and once it was determined that respondents did report interactions with one another, each respondent's ego-level metrics were calculated by entering the reported interaction levels into the Excel add-in, NodeXL [37]. With the overarching goal of identifying faculty members to assist with the diffusion of HIPs based on social interaction patterns, the following metrics were calculated: in-degree, out-degree, betweenness centrality, and closeness centrality. In-degree measures the extent to which an individual is identified as the target of a respondent's teaching interaction. Out-degree measures the extent to which an individual identifies others as the target of their teaching interaction. Betweenness centrality measures the extent to which an actor has the structural position to be able to make connections between two otherwise unconnected alters to help foster teaching interactions. Closeness centrality measures the extent to which an actor has a structural position to be able to facilitate quick communication throughout the network and help spread awareness of teaching practices.

With the intent to parse out the different levels of interaction, two subsets of these metrics were calculated. The first subset combined the ego metrics for the three interaction levels, 2, 3, and 4, known as the all-interaction network. Seeking to determine a difference in the interaction levels based on active collaboration, the second subset only included levels 3 and 4, known as the collaboration interaction network.

Homophily was also calculated to determine the propensity of individuals from each quartile to discuss teaching with others of similar TPI scores. Homophily measures the tendency of actors within a network to interact with others that have similar characteristics more frequently than interacting with others that have different characteristics and the metric is known as an H indicator [38]. For this study, the characteristic used to determine homophily was the TPI score. The overall network was divided into four distinct networks based on the TPI quartile. Each of the four quartiles received an H indicator to determine the propensity of actors in those quartiles to interact more frequently with one another, as opposed to actors with different TPI scores. Homophily scores were only calculated for all-interactions networks due to the small numbers of collaborative interactions.

The all-interaction network generated through the responses led to the inclusion of 43 faculty members with 108 interactions. The collaboration interactions network included 25 faculty members with 24 interactions. Various network metrics were correlated to TPI score to address H1: respondents with relatively higher TPI scores will demonstrate relatively higher ego metrics. Kendall's Tau correlations were run to determine relationships between the TPI scores and each of the ego-level metrics for each network subset. There were no significant correlations found between the TPI scores and the two subsets of ego-level metrics, as shown in Table 4.

**Table 4.** Kendall's Tau correlations between ego-level metrics and TPI scores for all-interactions network and collaboration interaction network. No significant correlations were found ($p > 0.05$ for all data).

| Ego-Level Metric | All Interaction Network | Collaboration Interaction Network |
|---|---|---|
| In-Degree | 0.25 | 0.056 |
| Out-Degree | −0.081 | −0.120 |
| Betweenness Centrality | −0.012 | −0.064 |
| Closeness Centrality | 0.159 | 0.274 |

An additional metric, known as a collegiality score, was calculated to further examine the network characteristics of the faculty members and their possible relationship to implementing high-impact practices. The collegiality score combines the betweenness centrality and in-degree metrics to determine individuals that are in positions to not only connect otherwise unconnected colleagues, but are also well-connected themselves [32]. These individuals possess network characteristics that can facilitate the diffusion of high-impact practices. Figure 3 shows the TPI score as a function of the collegiality metric. A linear

model fit to the data (solid line) had slope $m = 0.009$ and coefficient of determination $R^2 = -0.020$, indicating that the variance in TPI score is not at all explained by the collegiality metric. Similar to other ego-level metrics, this suggests that there is no correlation between social network interactions and levels of HIP usage within a course, falsifying H1.

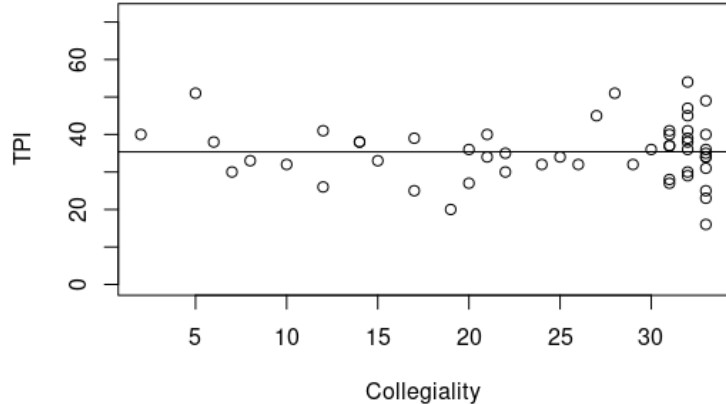

**Figure 3.** TPI score as a function of collegiality metric. The solid line is the linear model with slope $m = 0.0009$ and $R^2 = -0.020$, indicating no correlation between TPI score and faculty collegiality.

H indicators were calculated for each quartile to address H2: respondents with similar TPI scores will more frequently interact with one another than with individuals with different TPI scores. H indicator values range from $-1$ to $+1$. An H value less than 0 indicates a bias toward homophilic pairs with an increasing bias the farther from 0; an H value equal to 0 indicates no bias toward either homophilic or heterogenous pairs; and an H value greater than 0 indicates a bias toward heterogeneous pairs with an increasing bias, the farther from 0. Quartile 1's network demonstrated the most neutral H value indicating that actors with the lowest quartile score have no tendency toward interacting with others that have similar TPI scores or with others that have different TPI scores. Quartiles 2 and 3 also demonstrated neutral tendencies whereas Quartile 4 scored an H value farthest from 0 and positive, which demonstrates a slight bias in homophilic pairs over heterogeneous pairs. Results are found in Table 5 along with a sociogram of interactions between the quartiles in Figure 4. Other than a slight heterophilic pair bias for Quartile 4, respondents with similar TPI scores showed no preference for interactions within their group, falsifying H2.

**Table 5.** H indicator for each quartile with explanation.

| Quartile | H Indicator | Explanation |
| --- | --- | --- |
| 1 | −0.01 | No bias |
| 2 | 0.03 | No bias |
| 3 | −0.07 | No bias |
| 4 | 0.25 | Slight heterophilic pair bias |

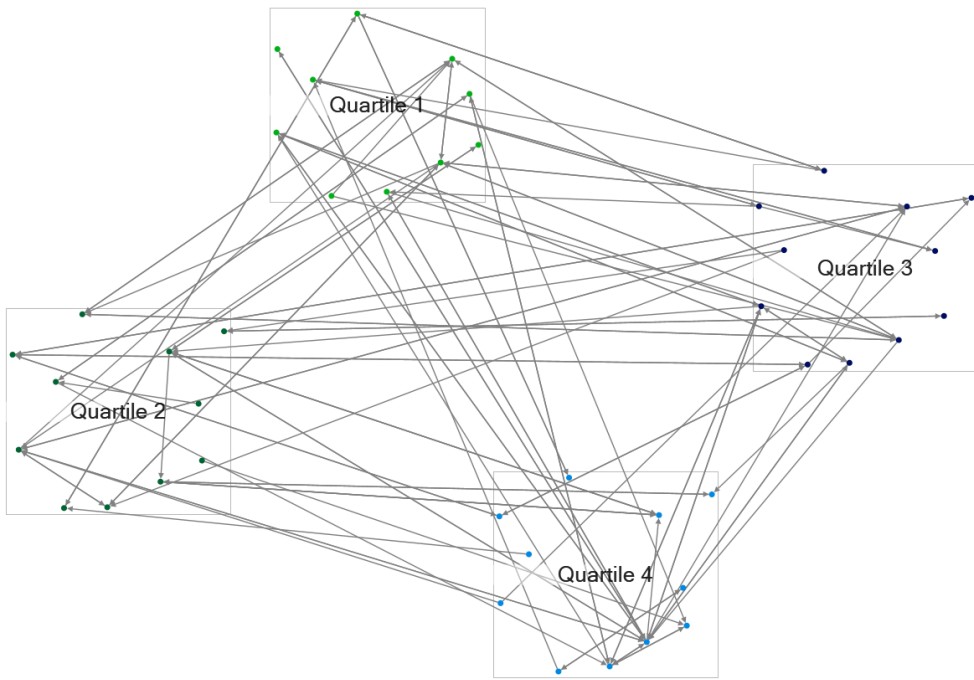

**Figure 4.** Sociogram of inter- and intra-quartile interactions. Each colored dot represents a network actor, with actors grouped by TPI quartile (represented by different colored dots and grouped into the boxes shown and labeled.

## 4. Discussion

The overarching research question for this study was as follows: How does a faculty member's implementation of HIPs relate to their teaching network? Based on the literature connecting SNA metrics with respect to teaching network positioning and use of HIPs in the classroom, we hypothesized that (1) respondents with higher TPI scores would demonstrate higher ego metrics, and (2) respondents with similar TPI scores would more frequently interact with one another. Both hypotheses were falsified by the study results. However, the falsification of these hypotheses by this study still sheds significant light on the research question. Although the literature suggests that a strong social network is a necessary condition of instructional change within an institution, our work suggests that it is not a sufficient condition. Teaching interactions among faculty are not enough on their own to diffuse the use of HIPs from high-users to low-users, even when significant interaction occurs across the two groups.

The absence of any significant correlations between faculty members' TPI scores and all the ego-level network metrics echo the results of Middleton et al. [11]. They conducted similar research that used SNA mixed methods to investigate the relationship between post-secondary faculty members' self-reported learner-centered attitudes scores and network metrics. They used the Approaches to Teaching Inventory (ATI) survey and in-degree and out-degree network metrics. Their results also yielded no significant relationships. However, in the same study, the researchers also incorporated a classroom observation protocol known as the Reformed Teaching Observation Protocol (RTOP), and several correlations between RTOP and network metrics were significant.

The similarities and differences between our current study and this previous study promote reflections for further studies. First, it is interesting that the self-report surveys, regardless of type, did not correlate with network metrics in either study. While the small sample size of both studies greatly limits any generalizability, the fact that both studies, conducted entirely separately from one another, yielded no significant correlations does necessitate attention.

There are many limitations with self-report surveys such as the TPI and the ATI, as well as SNA surveys. These limitations include social desirability bias, reference bias,

and introspective ability [39]. Social desirability bias occurs when individuals rate themselves "higher" hoping to appear more socially desirable, and this could have occurred in both the self-report instructional surveys as well as the SNA portion of the surveys. Reference bias occurs when individuals interpret rating scales differently, and again, this could have occurred for both the instructional portion, as well as the SNA portion of the surveys. Finally, introspective ability varies from person to person and some are able to rate themselves more objectively than others.

It should also be noted that the specifics of faculty interactions and discussions were not determined in either our study or the Middleton study [11]. For example, although faculty members may actively discuss teaching a course with other instructors, the SNA survey did not capture the content of those discussions. Discussions could have centered on content and curriculum issues instead of on specific practices. Or, specific practices could have been discussed, but not necessarily HIPs as outlined in this study.

There is some evidence within our results that interactions occurring across our population were more surface level. For the respondents to have been directed to the SNA portion of the survey, they had to respond within the TPI portion that they discussed teaching with their colleagues. Even though most respondents self-selected that they talked to colleagues regarding teaching, the depth of the conversations, on average, seemed to be relatively surface level considering the majority of the interactions did not include any type of collaborative activity centered around teaching.

Although many studies in both K–12 and higher education suggest that a strong social network is a necessary condition of instructional change within an institution, our work suggests that it is not a sufficient condition. This has implications for the diffusion of HIPs specifically and institutional change generally. The findings suggest that individuals, as well as institutional leaders, should use existing networks more strategically to diffuse HIPs. However, engaging individuals that possess both structural positions and pedagogical knowledge may be needed to help strategically diffuse HIPs in their own networks, with institutional support and guidance most likely also required to ensure deeper and more focused interactions.

At the time of this manuscript, further qualitative research is being conducted that is seeking to uncover the specific content of the teaching discussions happening within networks. Researchers hope that uncovering these specifics, across TPI scores as well as departments, will provide further insights into what is being discussed regarding teaching, who faculty members talk to, and the context behind their relationships and motivations. This knowledge should provide some contextual knowledge that is currently lacking in the research which will help researchers and institutional leaders better understand how to effectively activate these networks to help diffuse HIPs.

**Author Contributions:** Conceptualization, T.R., C.M., J.A.P. and S.E.; methodology, T.R.; formal analysis, T.R. and C.M.; investigation, T.R. and C.M.; resources, C.M.; data curation, T.R. and C.M.; writing—original draft preparation, T.R. and C.M.; writing—review and editing, J.A.P. and S.E.; project administration, C.M.; funding acquisition, C.M., J.A.P. and S.E. All authors have read and agreed to the published version of the manuscript.

**Funding:** This research was funded by the USA National Science Foundation Directorate for Undergraduate Education [REDACTED].

**Institutional Review Board Statement:** The study was conducted in accordance with the Declaration of Helsinki, and approved by the Institutional Review Board of [REDACTED].

**Informed Consent Statement:** Informed consent was obtained from all subjects.

**Data Availability Statement:** The data presented in this study are available on request from the corresponding author. The raw data are not publicly available in accordance with the IRB protocol.

**Acknowledgments:** The authors would like to acknowledge the administrative support of the University of Nebraska Omaha STEM TRAIL Center.

**Conflicts of Interest:** The authors declare no conflict of interest.

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
