# Peer review of "Barriers to Change: Social Network Interactions Not Sufficient for Diffusion of High-Impact Practices in STEM Teaching"

_education, doi:10.3390/educsci12080512_

Round 1
Reviewer 1 Report
This is an excellent research and presented well. I strongly recommend the following aspects considered and changed:
1. Title should reflect with HIP, TPI and social norm in terms of STEM education- make it catchy rather than simply focusing a relationship.
2. Before section 2.3 there is a need of a conceptual framework focused with key elements leading to HP1 & 2. This would provide a stronger picture of the research.
3. Before discussions it is better to provide a section of key findings in dot points and and some of the highlights, and this will make discussions reader friendly and logical.
Reviewer 2 Report
The authors did a great effort to present the relationship between social interactions an HIPs, with advanced and strong statistics. Multiple improvements could be emerged to increase the quality of this manuscript.
-The authors used High impact practices (HIPs) and High impact teaching practices (HIPs). It is consistent to have one expression.
-In the introduction: authors could present STEM as learning strategy, a set of teaching activities and a philosophy of improvement.
-Table (1): Practices that Support Learning and Teaching. Teaching is likely to be added to the table title.
-As the study conducted among social science courses, it is worth noting to mention STEAM as STEAM is closer to social sciences more than STEM.
-The study focuses on interactions and this implies to support the study with qualitative approach as observation or an interview.
-More discussion are needed to clarify the relationship between interactions and HIPs.
-The authors are encouraged to write some limitations that may encountered them during the study.
Round 2
Reviewer 2 Report
Dear authors
Thank you for your hard work and detailed response.